# Effects of Green Light on Elongation Do Not Interact with Far-Red, Unless the Phytochrome Photostationary State (PSS) Changes in Tomato

**DOI:** 10.3390/biology11010151

**Published:** 2022-01-17

**Authors:** Xue Zhang, Ep Heuvelink, Michaela Melegkou, Xin Yuan, Weijie Jiang, Leo F. M. Marcelis

**Affiliations:** 1Key Laboratory of Horticultural Crops Genetic Improvement (Ministry of Agriculture), Institute of Vegetables and Flowers, Chinese Academy of Agricultural Sciences, Beijing 100081, China; zhangxue19910605@foxmail.com; 2Horticulture and Product Physiology Group, Wageningen University, P.O. Box 16, 6700 AA Wageningen, The Netherlands; ep.heuvelink@wur.nl (E.H.); michaela.melegkou@syngenta.com (M.M.); xin.yuan@wur.nl (X.Y.)

**Keywords:** light spectrum, photoreceptor, *Solanum lycopersicum*, shade avoidance

## Abstract

**Simple Summary:**

This paper focuses on the role of phytochromes (phys) in the interaction between green light and far-red light effects on “shade avoidance syndrome”. We grew wild type and *phy* mutants of tomato under a set of light conditions with different combinations of green, blue, red, and far-red light. Partial (20%) replacement of red/blue by green light in the absence of far-red light hardly affected the tomato plant morphology. However, when the spectrum contained far-red light, partially replacing red/blue by green light resulted in more elongation, which was associated with a lower phytochrome photostationary state (PSS) value. There was no effect of partial substitution of red/blue with green light when the PSS was kept constant. Thus, this study has revealed an interaction between green and far-red light effects on elongation unless PSS was kept constant. Green light was often a bit neglected in photobiology, but now an increasing number of researchers are realizing that green light deserves more attention. This study advances the understanding of light quality and plant growth and finding the optimal spectrum when growing plants under LED lighting in controlled environment agriculture.

**Abstract:**

Green light (G) could trigger a “shade avoidance syndrome” (SAS) similarly to far-red light. We aimed to test the hypothesis that G interacts with far-red light to induce SAS, with this interaction mediated by phytochromes (phys). The tomato (*Solanum lycopersicum* cv. Moneymaker) wild-type (WT) and *phyA*, *phyB1B2*, and *phyAB1B2* mutants were grown in a climate room with or without 30 µmol m^−2^ s^−1^ G on red/blue and red/blue/far-red backgrounds, maintaining the same photosynthetically active radiation (400–700 nm) of 150 µmol m^−2^ s^−1^ and red/blue ratio of 3. G hardly affected the dry mass accumulation or leaf area of WT, *phyA*, and *phyB1B2* with or without far-red light. A lower phytochrome photostationary state (PSS) by adding far-red light significantly increased the total dry mass by enhancing the leaf area in WT plants but not in *phy* mutants. When the background light did not contain far-red light, partially replacing red/blue with G did not significantly affect stem elongation. However, when the background light contained far-red light, partially replacing red/blue with G enhanced elongation only when associated with a decrease in PSS, indicating that G interacts with far-red light on elongation only when the PSS changes.

## 1. Introduction

Plants grown under a canopy show “shade avoidance syndrome” (SAS), which is characterized by stem elongation, leaf hyponasty, and an increased specific leaf area (SLA) to attenuate shading by neighboring plants [1]. SAS is induced not only by a low total light intensity, but also by changes in spectral composition. Although a low red to far-red light ratio (R/FR ratio) is considered as the most important change in spectral composition responsible for SAS, reduced blue light (B) can also evoke SAS [2,3]. Moreover, increasing the green light (G) fraction in a background of R or B has been shown to activate some of the same responses that would be seen in FR light-induced SAS [4,5]. Wang et al. grew wild-type *Arabidopsis* plants with a low and high R/FR ratio with or without additional G, showing that additional G augmented the FR effects on petiole elongation [6]. An increased proportion of FR to R is a key signal for SAS. However, the fraction of G is supposed to be increased under a canopy, as it is less absorbed by leaves than R or B. G may be an additional signal for activating shading responses, enabling plants to adapt their development to a low-light environment under a canopy [7]. 

SAS is regulated by phytochromes (phys), as well as cryptochromes (crys), as SAS involves not only the responses to R and FR, but also the ratio of B to G [8,9,10]. Phys regulates SAS through two photo-interconvertible forms: the inactive (or red-absorbing (P_r_)) form and the active (or far-red-absorbing (P_fr_)) form [11]. The fraction of active phytochrome (P_fr_/P_tot_, P_tot_ = P_r_ + P_fr_) is called the phytochrome photostationary state (PSS) [12]. Five phy proteins (phyA to E) have been found in *Arabidopsis*, which are generally categorized into two groups: Light-labile type I (phyA) and light-stable type II (other phys) [13]. Type II phys are the prevalent phys of light-grown plants [14]. SAS induced by a low R/FR ratio (<1) is mostly mediated by the phyB photoreceptor, with phyD and phyE playing a minor role in *Arabidopsis* [15]. A high R/FR ratio activates phyB, resulting in repressing shade avoidance responses and resulting in a strong degradation of phyA, whereas a low R/FR ratio inactivates phyB but triggers phyA-mediated responses [11]. PhyA can make a direct contribution to shade responses, which becomes obvious in the phyB mutant background [16]. 

Two transcription factors, *PHYTOCHROME INTERACTING FACTOR 4* (*PIF4*) and *PIF5*, physically interact with the active form of phyB to regulate shade avoidance responses [17,18]. The *pif4* and *pif5* mutants both fail to respond to G and FR, indicating that *PIF4* and *PIF5* are convergence points for FR and G signals [7]. 

G likely affects plants mediated by crys, the photoreceptors of B and UV-A light [19]. There is a debate about the involvement of phys in G-induced responses. G increases hypocotyl elongation, which has been suggested to be mediated by crys rather than phyA or phyB [20,21]. A recent meta-analysis suggested that the shorter G wavelengths complement while the longer G wavelengths antagonize B-induced responses, either through directly repressing cryptochrome signaling or by a phytochrome-dependent mechanism [22]. This study aimed to investigate the possible interaction between G and FR and the role of phys in this interaction. We hypothesized that G interacts with FR to induce SAS, as mediated by phys. Our experiments were conducted in a climate chamber, where the effects of G (525 nm) were examined by partial (20%) substitution of background light (red/blue, RB) with G in the presence or absence of FR. In this study, we used tomato (*Solanum lycopersicum*) because it is the physiological and molecular model crop for fruit-bearing plants, and it has a day-neutral response of photoperiodic flowering [23]. The tomato phytochrome family also consists of five genes, *PHYA*, *PHYE* (*PHYC* in *Arabidopsis*), and three members of the *PHYB* subfamily, *PHYE* and two paralogs of *PHYB* (*PHYB1* and *PHYB2*) [24]. To understand the involvement of phys, we used three phys-deficient genotypes. In contrast to a number of earlier studies on G, we kept the photosynthetically active radiation (PAR; 400–700 nm) and R/B ratio the same when replacing 20% RB with G. The findings can make an important contribution to the field of regulating the spectrum when growing plants under LED lighting in controlled environment agriculture.

## 2. Materials and Methods

### 2.1. Plant Cultivation and Light Treatments

The tomato seeds (*Solanum lycopersicum* cv. Moneymaker) were acquired from the Tomato Genetic Resource Center (UC Davis, USA), including wild-type (WT) and mutants *phyA* (lack of phyA), *phyB1B2* (lack of phyB1 and B2), and *phyAB1B2* (lack of phyA, B1, and B2). Details on the set-up of the experiment and cultivation practices can be found in Zhang et al. [25], as these were similar, but light treatments and genotypes were different. Seeds were germinated in darkness. After three days, they were transferred to 150 μmol m^–2^ s^–1^ white light-emitting diode (LED) light (GreenPower, Philips, Eindhoven, the Netherlands). The temperature was 22 °C during the day and 18 °C during the night; the relative air humidity was 70% and the photoperiod was 16 h. 

Ten days after sowing, the light treatments started. Pots with expanded clay grit were placed in 60 cm × 40 cm × 7.5 cm containers, where a constant layer of standard tomato nutrient solution (EC = 2.0) was retained. The nutrient solution was refreshed every two to three days.

There were two series of light treatments, as described in Table 1 and Table 2. In Experiment 1 (Exp1), the treatments consisted of RB (R/B ratio = 3) with or without FR and/or G (Table 1). Plants were grown under different light treatments of the same total PAR (around 150 μmol m^−2^ s^−1^ measured just above the plants). Both the R and B intensity were reduced when replaced with G, to keep the same R/B ratio as in the treatment without G. The PSS values changed when partially replacing RB with G in the presence of FR. Therefore, we conducted Experiment 2 (Exp2), where the PSS values were kept constant by reducing the FR intensity when G was partially replaced with RBFR (R/B ratio = 3) (Table 2).

Light was supplied by four colors of narrow-band LEDs of various peaks: 447 nm (B; GreenPower, Philips, Eindhoven, The Netherlands), 525 nm (G; Lumileds, San Jose, CA, USA), 667 nm (R; GreenPower, Philips, Eindhoven, The Netherlands), and 730 nm (FR; GreenPower, Philips, Eindhoven, The Netherlands) (Figure A1). The photon flux density (PFD; 400–800 nm) was measured with an Apogee^®^ Spectroradiometer SS-110, and the fractions of B (400–500 nm), G (500–600 nm), R (600–700 nm), and FR (700–800 nm) light in each treatment were calculated. The PSS was calculated according to the method of Sager et al. [12]. 

### 2.2. Measurements

The plants were destructively measured 3 weeks after the start of the treatments in both Exp1 and Exp2. The stem length was measured from the upper surface of the grit up to the shoot tip. The total leaf area (without cotyledons) was measured by a leaf area meter (model LI-3000; LI-COR, Lincoln, NE, USA), and the leaf number was measured simultaneously. The dry mass was determined by drying the leaves, stems, and roots in a ventilated oven (105 °C, 24 h). Based on the measured leaf area and the leaf dry mass, the SLA (square meter of leaf area per gram of leaf dry mass) was calculated. 

### 2.3. Statistical Set-Up and Analysis 

Exp1 was conducted five times, while Exp2 was conducted four times. Each time represented one block. In each block of both experiments, three individual plants were measured for each light treatment × genotype combination (nine plants for stem length); hence, each combination in total had 15 replicate plants in Exp1 and 12 in Exp2 (45 in Exp1 and 36 in Exp2 for stem length). The experiment was designed with a split plot arrangement, where light treatments were applied to the whole plots and genotypes to the subplots. Data were statistically analyzed by analysis of variance (ANOVA) using Genstat Software, version 19.0. Residual’s normality was tested by a Shapiro–Wilk test at *p* = 0.05. Although the residuals of stem length in Exp1 could not be considered normal (*p* = 0.026), we did not remove any outliers, nor did we apply data transformation, since the residual plot showed no clear problem with normality. The means were separated using the Fisher’s unprotected LSD test at *p* = 0.05. In the case of no significant interaction by the F-test, we also compared the interaction averages (light treatment × genotype) by this unprotected test.

## 3. Results

### 3.1. Effects of Green and Far-Red Light on Elongation

Figure 1 shows representative plants from 16 combinations of four light treatments and four genotypes in Exp1. The stem length of all genotypes was significantly increased when adding FR to RB or to RBG (Figure 2). Partially (20%) replacing RB with G did not significantly affect the stem length when the spectrum did not contain FR (Figure 2). However, when there was FR in the spectrum, partially replacing RB with G (RBFRG compared to RBFR), which implies a lowering of the PSS value (Table 1), resulted in taller plants in the wild-type WT, *phyA*, and *phyAB1B2* but not in *phyB1B2*. On the contrary, when there was FR in the spectrum but the PSS values were kept constant (Table 2), partially replacing RB with G did not significantly affect the stem length in any genotypes (Figure 3). 

The *phyB1B2* mutant plants were the tallest among the four genotypes under all light treatments (Figure 2 and Figure 3). The *phyA* mutant plants showed a similar stem length to the plants of WT under RB and RBG, whereas they were more elongated compared to the WT plants in the presence of FR (only significant in Exp2). The *phyAB1B2* mutant plants showed significantly longer stems than WT under RB, whereas they were significantly shorter under RBFR (in both Exp1 and Exp2). The *phyAB1B2* mutant plants had shorter stems than *phyA* under RBFR and RBFRG (only significant in Exp2), but no differences were found between these two genotypes under RB and RBG.

### 3.2. Effects of Green and Far-Red Light on Dry Mass, Shoot/Root Ratio, and Leaf Area 

Additional FR increased the total dry mass of WT plants but did not affect that of *phy* mutants in most light treatments (Figure 4). Partially replacing RB with G hardly affected the dry mass both in the presence and absence of FR (Figure 4 and Figure A2). All *phy* mutants had lower dry mass than WT under RBFR and RBFRG, though this was not always statistically significant (Exp1, Figure 4). However, there was almost no effect of G in any of the genotypes when the PSS was kept constant (Exp2, Figure A2). 

Additional FR increased the shoot/root ratio in all genotypes, though this was not always statistically significant (Figure 5). Partially replacing RB with G had almost no effect on the shoot/root ratio.

Additional FR increased the leaf area of WT, whereas it hardly affected the leaf area of the mutants (Figure 6). Partially replacing RB with G increased the leaf area of *phyAB1B2* when the spectrum contained FR, associated with a decreased PSS value (Figure 3B and Figure 6D). Compared to the wild-type plants, the *phyA* and *phyB1B2* mutant plants had a decreased leaf area in the presence of FR, while the *phyAB1B2* showed a decreased leaf area in both the presence and absence of FR, but this was not always significant (Exp1, Figure 6). The *phyAB1B2* mutant did not significantly diverge in leaf area from *phyB1B2* under RBFR and RBFRG, whereas it had a lower leaf area under RB and RBG (Figure 3B and Figure 6). The leaf number and SLA hardly differed between genotypes or light treatments (Figure A3 and Figure A4).

## 4. Discussion

### 4.1. No Interaction between Green and Far-Red Light on Elongation Unless the PSS Changes 

Elongation increased when partially replacing RB with G when there was also FR in the spectrum (Figure 1 and Figure 2). However, G did not affect elongation when the PSS values remained unchanged (Figure 3). Green light, as such, hardly affects the PSS value, but replacing 20% of the RB background with G may result in changed PSS values due to the fact that the ratio of FR to the other wavelengths changes. Therefore, a lower PSS value is a likely explanation for the increased elongation when partially replacing RB with G when there was also FR in the spectrum (Figure 1 and Figure 2). So, based on our results, we conclude that there is no effect of G on elongation unless PSS changes. However, one study found that G increases elongation despite R/FR (hence PSS) being kept constant [5], while in another study, where R/FR (hence PSS) dropped when G was added, no effect of G was found on elongation [26]. 

In *Arabidopsis*, phyB is the dominant phytochrome in the control of internode elongation, and phyD and phyE play minor and/or redundant roles [27]. However, consistent with the results of Weller et al. [28], our data showed that the stem length of *phyAB1B2* mutant plants could still respond to additional FR and a decrease in PSS when replacing RB with G, indicating that at least one other phytochrome (phyE or phyF) is active in mediating stem elongation in tomato. This is also supported by Schrager-Lavell et al. and Li et al., revealing a role of tomato phyE in regulating the shade avoidance response and a role of Maize ZmphyCs (high sequence similarity with tomato phyF) in regulating plant height, respectively [24,29]. This suggests that no phys are redundant, although their functions may partly overlap.

### 4.2. Green Light Had No Effect on Total Dry Mass, Except in the phyAB1B2 Mutant When Interacting with FR

Partially replacing RB with G hardly affected the total dry mass, shoot/root ratio, or leaf area in the WT and *phyA* and *phyB1B2* mutant plants in both the presence and absence of FR (Figure 4, Figure 5 and Figure 6). This is in line with our previous research and that of Virsile et al. when growing lettuce under a mixture of R, B, and FR, as these authors did not observe significant effects on lettuce biomass when partially replacing R with G [25,26]. However, adding 520 nm G to RBFR significantly increased the dry mass and leaf area of cucumber plants [30], as well as young tomato plants [31]. The discrepancies might be due to an increased total light intensity when adding G in those studies. 

In contrast to other genotypes, the leaf area and total dry mass of the *phyAB1B2* mutant were significantly increased by G when the spectrum contained FR (Figure 4D and Figure 6D). However, it is not easy to interpret why only the *phyAB1B2* mutant had such a response. As discussed above, we propose that phyE and/or phyF may play a role in this response.

### 4.3. Far-Red Light Increased the Wild-Type Dry Mass Resulting from an Enhanced Leaf Area, Whereas It Decreased the phyB1B2 Mutant’s Leaf Area 

Additional FR significantly increased the dry mass and leaf area (Figure 4A and Figure 6A) of WT plants, which is in line with other reports [32,33]. Additional FR increased the plant dry mass, resulting from increased light absorption as a consequence of an increased leaf area [32]. Meanwhile, additional FR increased the shoot/root ratio (Figure 5). We conclude that FR stimulates dry matter partitioning to the shoot and enlarges the leaf area and stem length, resulting in an increase in plant dry matter accumulation. Furthermore, the dry mass and leaf area of the *phyA* and *phyAB1B2* mutant plants were hardly affected by FR, indicating the role of phys in these FR effects. 

In contrast to the wild-type plants, the leaf area of the *phyB1B2* mutant was significantly decreased by additional FR and was significantly smaller than WT when the spectrum contained FR (Figure 6B,C). Similar to our results, *Arabidopsis* displayed a reduced leaf area in the *phyB* and *phyBphyD* mutants, suggesting that this effect is dependent on the activity of phyB and phyD (high sequence similarity with tomato phyB1 and phyB2, respectively) [34,35,36]. This inhibitory effect of FR on *phyB1B2* leaf expansion may in part result from strong competition for resources with the stem [37], as *phyB1B2* had significantly longer stems than the other genotypes, explaining our findings that additional FR strongly increased the shoot/root ratio of *phyB1B2* but did not affect its total dry mass (Figure 5 and Figure 6).

## 5. Conclusions

Partial (20%) replacement of RB by G in the absence of FR hardly affected the tomato plant morphology. However, when the spectrum contained FR, partially replacing RB by G resulted in greater elongation, which was associated with a lower PSS value. There was no effect of partial replacement of RB by G when the PSS was kept constant. Thus, we conclude that there is no interaction between G and FR on elongation unless PSS changes. Increased elongation by additional FR in the *phyAB1B2* mutant outlines a possible role for *PHYE* and/or *PHYF* in the stem elongation response to FR. FR stimulates dry matter partitioning to the shoot and enlarges the leaf area and stem length, resulting in an increase in plant dry matter accumulation, but not in *phy* mutants, confirming the involvement of phys.

## Figures and Tables

**Figure 1 biology-11-00151-f001:**
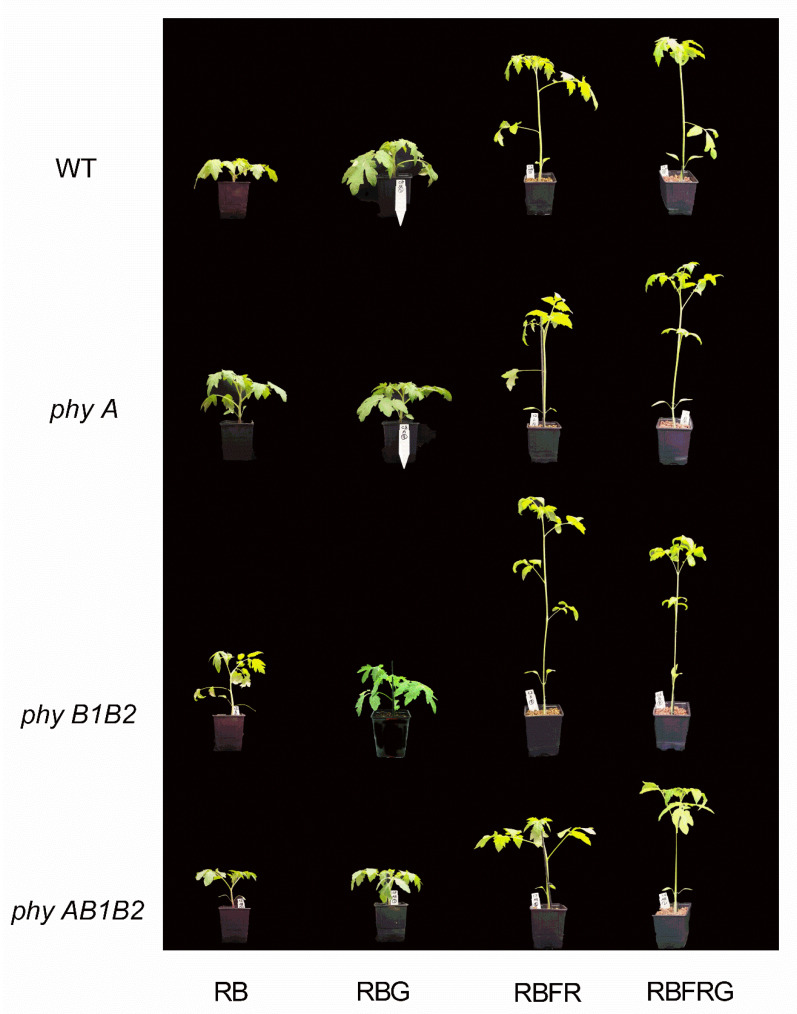
Representative pictures showing the effect of partial (20%) substitution of red/blue (RB; 3:1) with green (G) light in the presence or absence of far-red light (FR) on tomato phenotypes 21 days after treatment. Four genotypes were studied: WT (Moneymaker, wild-type), *phyA* (phytochrome A-deficient), *phyB1B2* (phytochrome B1- and B2-deficient), and *phyAB1B2* (phytochrome A-, B1-, and B2-deficient) (Exp1).

**Figure 2 biology-11-00151-f002:**
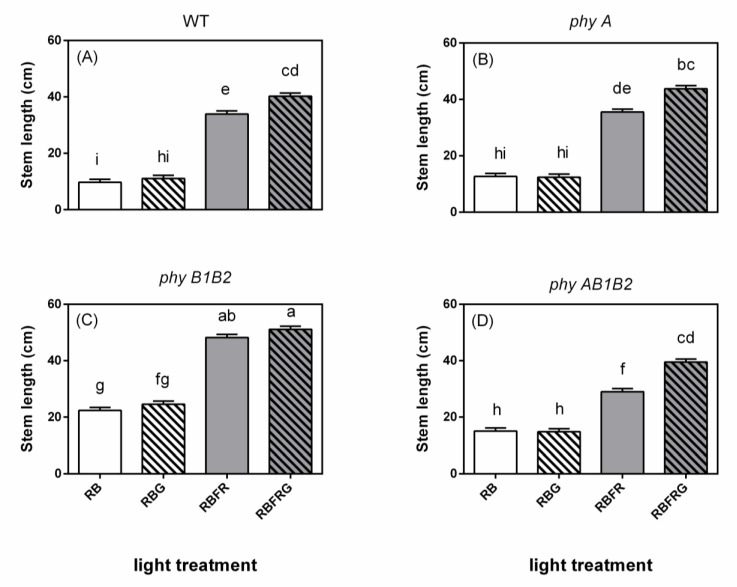
Effect of partial (20%) substitution of red/blue (RB; 3:1) with green (G) light in the presence or absence of far-red light (FR) on the stem length of tomato plants 21 days after treatment. Four genotypes were studied: (**A**) WT (Moneymaker, wild-type), (**B**) *phyA* (phytochrome A-deficient), (**C**) *phyB1B2* (phytochrome B1- and B2-deficient), and (**D**) *phyAB1B2* (phytochrome A-, B1-, and B2-deficient). Light treatment and genotype significantly interacted (*p* = 0.034). Lower case letters indicate significant differences among the combinations of light treatment and genotype (*p* = 0.05), thus also allowing comparisons between panels (**A**–**D**). Data presented are averages with the SE based on 5 blocks (*n* = 5), each containing 9 replicate plants (Exp1).

**Figure 3 biology-11-00151-f003:**
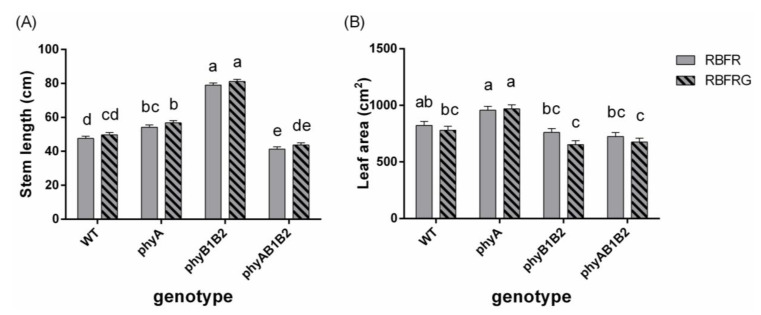
Effects of partial (20%) substitution of red/blue (RB; 3:1) with green (G) light while keeping the PSS constant on (**A**) the stem length and (**B**) plant leaf area of tomato plants 21 days after treatment. Four genotypes were studied: WT (Moneymaker, wild-type), *phyA* (phytochrome A-deficient), *phyB1B2* (phytochrome B1- and B2-deficient), and *phyAB1B2* (phytochrome A-, B1-, and B2-deficient). Light treatment and genotype did not significantly interact (for stem length, *p* = 0.999; for leaf area, *p* = 0.638). The effects of light treatment were not significant (for stem length, *p* = 0.075; for leaf area, *p* = 0.34), but the effects of genotype were significant (*p* < 0.001 for both parameters). Lower case letters indicate significant differences among the combinations of light treatment and genotype (*p* = 0.05). Data presented are averages, with the SE based on 4 blocks (*n* = 4), each containing 9 replicate plants for stem length and 3 replicate plants for leaf area (Exp2).

**Figure 4 biology-11-00151-f004:**
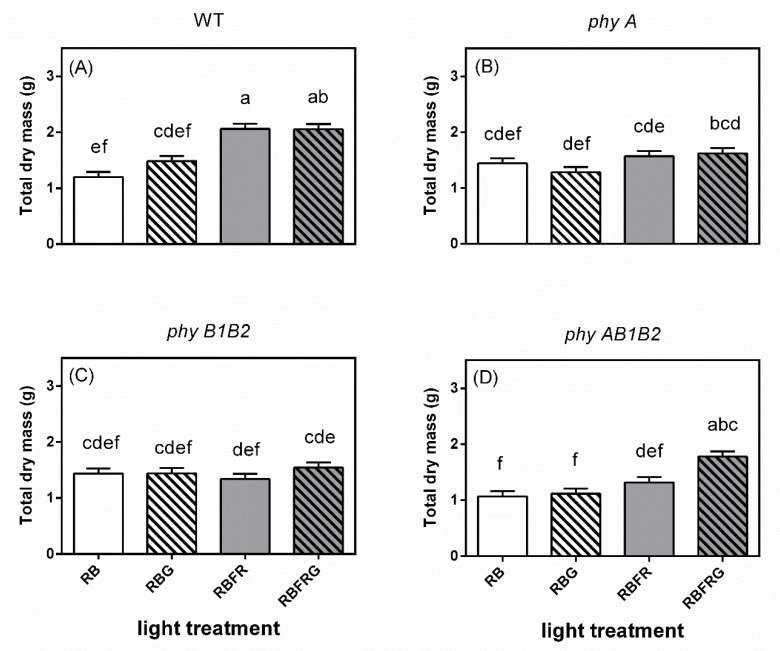
Effect of partial (20%) substitution of red/blue (RB; 3:1) with green (G) light in the presence or absence of far-red light (FR) on the total dry mass of tomato plants 21 days after treatment. Four genotypes were studied: (**A**) WT (Moneymaker, wild-type), (**B**) *phyA* (phytochrome A-deficient), (**C**) *phyB1B2* (phytochrome B1- and B2-deficient), and (**D**) *phyAB1B2* (phytochrome A-, B1-, and B2-deficient). Light treatment and genotype did not significantly interact (*p* = 0.082), although the effects of light treatment and genotype were significant (*p* < 0.001 and *p* = 0.01, respectively). Lower case letters indicate significant differences among the combinations of light treatment and genotype (*p* = 0.05), thus also allowing comparisons between panels (**A**–**D**). Data presented are averages, with the SE based on 5 blocks (*n* = 5), each containing 3 replicate plants (Exp1).

**Figure 5 biology-11-00151-f005:**
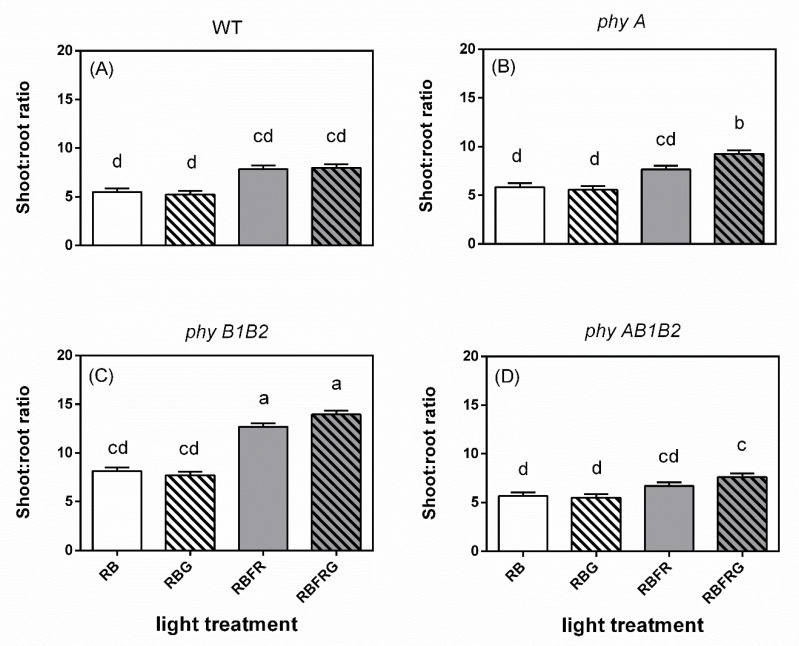
Effect of partial (20%) substitution of red/blue (RB; 3:1) with green (G) light in the presence or absence of far-red light (FR) on the shoot/root ratio of tomato plants 21 days after treatment. Four genotypes were studied: (**A**) WT (Moneymaker, wild-type), (**B**) *phyA* (phytochrome A-deficient), (**C**) *phyB1B2* (phytochrome B1- and B2-deficient), and (**D**) *phyAB1B2* (phytochrome A-, B1-, and B2-deficient). Light treatment and genotype significantly interacted (*p* = 0.001). Lower case letters indicate significant differences among the combinations of light treatment and genotype (*p* = 0.05), thus also allowing comparisons between panels (**A**–**D**). Data presented are averages, with the SE based on 5 blocks (*n* = 5), each containing 3 replicate plants (Exp1).

**Figure 6 biology-11-00151-f006:**
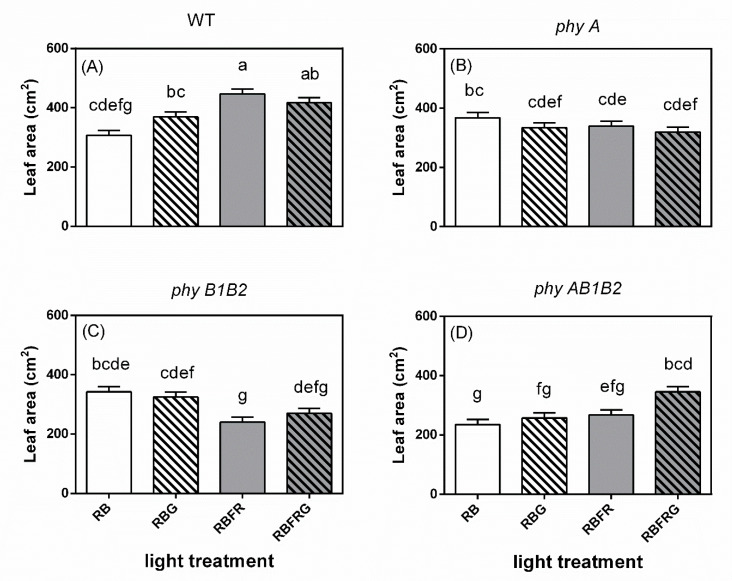
Effect of partial (20%) substitution of red/blue (RB; 3:1) with green (G) light in the presence or absence of far-red light (FR) on the leaf area of tomato plants 21 days after treatment. Four genotypes were studied: (**A**) WT (Moneymaker, wild-type), (**B**) *phyA* (phytochrome A-deficient), (**C**) *phyB1B2* (phytochrome B1- and B2-deficient), and (**D**) *phyAB1B2* (phytochrome A-, B1-, and B2-deficient). Light treatment and genotype significantly interacted (*p* = 0.003). Lower case letters indicate significant differences among the combinations of light treatment and genotype (*p* = 0.05), thus also allowing comparisons between panels (**A**–**D**). Data presented are averages, with the SE based on 5 blocks (*n* = 5), each containing 3 replicate plants (Exp1).

**Table 1 biology-11-00151-t001:** The measured total photosynthetically active radiation (PAR; 400–700 nm) and set-points photon flux density (PFD; 400–800 nm) of blue (B; 400–500 nm), green (G; 500–600 nm), red (R; 600–700 nm), and far-red (FR; 700–800 nm) light and the phytochrome photostationary state (PSS) for the four spectral treatments in Experiment 1.

Spectral Treatment	Light Intensity (μmol m^−2^ s^−1^)					
	Total PAR *	Red (R)	Blue (B)	Green (G)	Far-Red (FR)	PSS **
RB	150.44 ± 1.07	112.5	37.5			0.88
RBG	149.08 ± 2.02	90	30	30		0.88
RBFR	146.80 ± 1.53	112.5	37.5		100	0.73
RBFRG	150.63 ± 0.64	90	30	30	100	0.68

* Light was measured just above the plants. Average ± SEM, *n* = 5. The raw data are shown in Table A1. ** The PSS was calculated according to the method of Sager et al. [12].

**Table 2 biology-11-00151-t002:** The measured total PAR (400–700 nm) and set-points PFD (400–800 nm) of blue (B; 400–500 nm), green (G; 500–600 nm), red (R; 600–700 nm), and far-red (FR; 700–800 nm) light and the PSS for the two spectral treatments in Experiment 2.

Spectral Treatment	Light Intensity (μmol m^–2^ s^–1^)					
	Total PAR *	Red (R)	Blue (B)	Green (G)	Far-Red (FR)	PSS **
RBFR	144.06 ± 0.62	105	35		40	0.80
RBFRG	141.03 ± 0.88	90	30	30	34	0.80

* Light was measured just above the plants. Average ± SEM, *n* = 4. The raw data are shown in Table A2. ** The PSS was calculated according to the method of Sager et al. [12].

## Data Availability

The data presented in this study are available on request from the corresponding author. The data are not publicly available due to privacy.

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
