# Peer review of "Effects of Green Light on Elongation Do Not Interact with Far-Red, Unless the Phytochrome Photostationary State (PSS) Changes in Tomato"

_biology, 2022, doi:10.3390/biology11010151_

Round 1
Reviewer 1 Report
The manuscript by Zhang et al. describes effect of green light on growth and development in wild type and phytochrome mutant plants under various PSS states. However, this manuscript is not sufficiently described as pointed out below.
Authors should designate wild-type Moneymaker as “WT” but not “MM”, if all phytochrome mutants are in the Moneymaker background.
I do not know why authors used both red and blue light and change the light intensity of both light between RB and RBG (and between RBFR and RBFRG). Lowering blue light intensity could affect phytochrome and cryptochrome signaling pathways and thus make the interpretation more difficult.
Fig. 2-6 and A2-A4
The lines of bar chart and error bars are too thick. Most error bars cannot be recognized. Additionally, color of bar chart is strange. Purple color reminds me of UV light. Authors should make chart bar color and design much simpler; for example, RB: white, RBG: black, RBFR: gray, RBFRG: black and gray striped.
Page 2, lines 54-56
This sentence is open to misunderstanding. There is no evidence that phototropins and UVR8 regulate SAS. Conversely, shade indirectly enhances phototropin-mediated phototropism (Goyal et al. Curr. Biol. Vol. 26, pp. 3280-3287, 2016) and UV-B (and thus UVR8) inhibit SAS (Hayes et al. PNAS. Vol. 111, pp. 11894-11899, 2014; Tavridou et al. Plos Genet. Vol. 16, e1008797, 2020).
Page 2, lines 54-78
This part is highly complicated. Authors introduced the number and type of tomato phytochromes, the description of SAS and PIF transcription factors is based on studies of Arabidopsis. Authors should firstly introduce phytochrome and SAS by using Arabidopsis research and finally explain why authors used tomato and what kind of phytochromes existed in tomato (in lines 79-86).
Page 4, line 164
“phyB1B2” should be italicized.
Figure 2 and Figure 3
I do not know why authors changed the light composition for RBFR. The PSS in RBFR and RBFRG is different between Figure 2 and Figure 3. Thus, we cannot directly compare the results between Fig. 2 and Fig. 3.
The size and height of phyB1B2 were clearly larger under RBG than those under RB (Figure 1). However, the biomass phyB1B2 was comparable between RB and RBG in Figure 4, 5, and 6. Why?
Reviewer 2 Report
The following manuscript entitled “Effects of green light on elongation do not interact with far-red, unless the phytochrome photostationary state (PSS) changes” provides a detailed study on the role of phytochromes in the interaction effect between the green light and far-red light on “shade avoidance syndrome”. The manuscript is well written and thoroughly described, however, there are certain changes needed in the manuscript in order to make it acceptable as a publication. Moreover, there are some minor/major grammar and sentence checks required throughout the manuscript by some English language experts in this subject.
Title/Simple summary/Abstract:
Please clarify the type of the article, e.g., Research article? If the study is a Research, this Section must contain: Background/Aim - Materials and Methods - Results - Conclusion(s).... etc
The title should be revised. It should be concise and informative regarding your aims/objectives and it should identify if the study objects (plants/animals/humans)
Say something about the future prospects of your study at the end of this section as well.
Keywords:
keywords must be different from title words. Avoid repetition
Introduction:
Mention the required/recommended photoperiod of tomato at different growth stages especially at the reproductive phase.
Provide literature/information regarding the negative effect of photoperiod stress at different growth stages in tomato plants
Moreover, at the end of the introduction section, correlate your findings with the hypothesis and its future prospects, how the industries/researchers in this field can benefit from your achieved results.
Methodology:
The methodology of the proposed study is fine and well-designed. However, I think the paragraph in lines 109-119 should be mentioned in a tabular form as well, in order to better understand the treatments.
Results and Discussion:
This portion lacks scientific reasoning or justification of research results, incorporate it.
The discussions are a little too lengthy, which is a good thing but on the same note, it makes it a bit speculative. The authors should stick to the specific/solid justifications that are correlated with their developed arguments. Please use a more scientific style.
Moreover, there are some language and grammar mistakes in this section as well.
The references are not properly formatted. Please follow the author’s guidelines provided by the journal.
To sum up, this manuscript can find interest to the researchers and readers in this field. However, the above-mentioned minor changes should be addressed before its possible acceptance.
Good Luck!
Reviewer 3 Report
It is an interesting study of the influence of green, red, blue and far red light on tomato growth. The strong points are represented by the experimental design and by the synthetic presentation of the results. The conclusions are supported by the results obtained.
Specific recommendations:
L106-108 please specify how the PSS was kept constant (it can be deduced that by reducing the FR intensity but this must be specified).
L114 please specify briefly how you calculated PSS, this will facilitate the documentation activity of researchers who want to repeat the experiment.
The presentation of the statistical test in figures may mislead the reader due to the fact that it is generally customary for the comparison to be made only within a graph even if it is specified in the description of the figure how the test was performed.
Round 2
Reviewer 1 Report
Page 11, lines 296-297
This sentence is open to misunderstanding. Do authors mean that phyD, but not phyB, shows "high sequence similarity with tomato phyB1 and phyB2"? Probably, they want to say that "this effect is dependent on the activity of phyB and phyD, both of which have high sequence similarity with tomato phyB1 and phyB2"
Author Response
Dear Reviewer,
Thank you for your comment concerning our manuscript entitled “Effects of green light on elongation do not interact with far-red, unless the phytochrome photostationary state (PSS) changes in tomato”. This comment is valuable and helpful for revising and improving our manuscript. Revised portions are tracked and marked in green in the resubmitted manuscript.
Response to the reviewer’ comment:
Page 11, lines 296-297
This sentence is open to misunderstanding. Do authors mean that phyD, but not phyB, shows "high sequence similarity with tomato phyB1 and phyB2"? Probably, they want to say that "this effect is dependent on the activity of phyB and phyD, both of which have high sequence similarity with tomato phyB1 and phyB2".
Response: In accordance to the comment of the reviewer, we changed the sentence into “Similar to…, suggesting that… on the activity of phyB and phyD (high sequence similarity with tomato phyB1 and phyB2, respectively) [34-36].”
We appreciate this constructive comment of the referee which helped us to improve the manuscript.
Sincerely,
Prof. Dr. Leo Marcelis
Wageningen University